# Increasing Physical Activity in Empty Nest and Retired Populations Online: A Randomized Feasibility Trial Protocol

**DOI:** 10.3390/ijerph17103544

**Published:** 2020-05-19

**Authors:** Amy Cox, Ryan Rhodes

**Affiliations:** Behavioural Medicine Laboratory, Department of Education, University of Victoria, Victoria, BC V8W 3N4, Canada; rhodes@uvic.ca

**Keywords:** physical activity, retirement, empty nest, life course transitions, habit, identity, behavioral change

## Abstract

Despite the extensive evidence on the benefits of physical activity (PA) in older adults, including reduced risk of disease, mortality, falls, and cognitive and functional decline, most do not attain sufficient PA levels. Theoretical work suggests that behavioral change interventions are most effective during life transitions, and as such, a theory-based, online intervention tailored for recently retired and empty nest individuals could lend support for increasing levels of PA. The aim of this study is to examine the feasibility of the intervention and study procedures for a future controlled trial. This study has a randomized controlled trial design with an embedded qualitative and quantitative process evaluation. Participants are randomized at 1:1 between the intervention and waitlist controls. Potential participants are within six months of their final child leaving the familial home or within six months of retiring (self-defined), currently not meeting the Canadian PA guidelines, have no serious contraindications to exercise, and are residing in Victoria, British Columbia, Canada. Participants are recruited by online and print flyers as well as in-person at community events. The study aims to recruit 40 empty nest and 40 retired participants; half of each group received the intervention during the study period. The internet-delivered intervention is delivered over a 10-week period, comprising 10 modules addressing behavior change techniques associated with PA. Primary outcomes relate to recruitment, attrition, data collection, intervention delivery, and acceptability. Secondary behavioral outcomes are measured at baseline and post-treatment (10 weeks). Intervention-selected participants are invited to an optional qualitative exit interview. The results of this feasibility study will inform the planning of a randomized effectiveness trial, that will examine the behavior change, health-related fitness, and well-being outcomes by exploring how reflexive processes of habit and identity may bridge adoption and maintenance in behavioral adherence.

## 1. Introduction

Engaging in a healthy lifestyle at midlife can help deflect a number of aging-related declines, including the loss of physical and cognitive function, weakening of subjective well-being, quality of life, social support, connectedness, and a reduced capacity to live independently [1,2,3,4,5,6,7,8]. Regular physical activity (PA) is a lifestyle pillar upon which health may be supported in all these domains; therefore, engaging in PA has the potential to substantially increase an individual’s quality of life, while reducing reliance on health and social services as people age. For adults, 150 min of moderate to vigorous aerobic PA per week is recommended [9], and yet involvement in PA continues to be plagued with low participation rates across the lifespan [10,11].

While the onset of parenthood is clearly associated with significant declines in PA for adults [12], other life transitions, such as retirement and the end of child rearing, may serve to restructure individuals’ lives in ways that reduce barriers to regular PA. Retirement, for example, may eliminate work related pressures [13,14,15,16,17,18,19,20], while post-child rearing may alleviate family obligations [21,22,23,24,25]. As such, both life transitions may substantially reduce time constraints, making regular PA levels easier to achieve. Research generally supports the hypothesis that retired people have more time to allocate to PA leisure activities such as gardening, walking, and sport participation [26,27,28,29]. Though there has been a lack of sustained empirical enquiry into the activity patterns of parents entering the empty nest period, it is intuitive that leisure time would be inversely associated with the time spent performing child care [30].

Importantly, life course transitions may also impact upon habits and identity in ways that influence PA and sustained behavioral change. After the preliminary phase of behavioral initiation, the continuation of PA behavior is thought to be influenced by reflexive processes of habit (processes of learned stimulus–response association [31]) and identity (which incites to align self-concept and behavior enactment) [32], as maintenance constructs. These concepts are embedded within the conceptual framework employed in this study, the multi-process action control model, which suggests that habits and identity change may be of particular utility to these populations proposed [33,34]. Being related to habit and social roles, both retirement and empty nest phases may also lead to a reassessment of life goals and reframing of identity [35,36,37]. As such, both transitions may be a time to dislodge old habits and create new ones [38,39,40,41]. Thus far, research (mostly within travel mode choices [42,43,44]) supports the habit discontinuity hypothesis, which states that behavior change interventions are most effective when delivered in the context of life course changes, as habit strength may be weakened, or even broken, by a sufficiently large change in the context in which a behavior is performed [38].

While the habit formation concept has seen promising results in physical activity research, [31,45,46,47] to our knowledge, the habit discontinuity theory has not yet been utilized to explore the relationships that may exist between PA levels and retirement, or empty nest transitions. The paucity of research on promoting PA during these times of life transition points to the need to explore whether or not these transition points may be a uniquely effective “window of opportunity” for PA health interventions. Systematic reviews that have synthesized intervention data from the retirement life stage lend support for their effectiveness on PA and other health outcomes [48,49,50]. Yet, still relatively few randomized controlled trials (RCTs) have purposefully delivered behavioral health interventions to the recently retired [51,52,53] and to our knowledge, there are no published reports of PA interventions targeting empty nest population groups.

## 2. Study Objectives

Consistent with the objectives for feasibility studies, the primary aim of this study is to examine the feasibility of the trial methods, processes, and acceptability of a theory-based online physical activity intervention targeting recently empty nest and retired individuals for increased physical activity levels. Outcomes will be assessed relating to recruitment, retention, data collection, intervention delivery, and satisfaction/acceptability. Further to these objectives, we seek to evaluate participants’ experiences and opinions of the intervention, as a basis for refining a future RCT.

Research Questions:What are the recruitment and retention rates of the study, established by the number of participants who were identified, eligible, consented, randomized, completed the program, and followed-up 10 weeks after baseline?What are the most appropriate outcome measures for a future RCT, considering the acceptability, reliability, and data quality of the administered measures?How acceptable are the intervention and trial procedures for participants?

Secondary objectives include evaluating the self-reported moderate and vigorous PA (MVPA) levels, change of habits, identity, and other PA-relevant behavioral constructs at 10 weeks.

### 2.1. Rationale

In their (2018) systematic review of effectiveness in technology-based interventions promoting the mental health and wellbeing of people aged 65 and over, Forsman and colleagues point out that many of the studies in this field concentrate on supporting older adults in a fragile state and/or those living with chronic conditions; however, there is also a need to promote healthy aging in the general population [54]. They also note that very few studies in their review took the form of randomized controlled trials (RCTs) and conclude that that there is a lack of methodologically rigorous evaluations within this area. More generally, reviews of technology-based interventions on patient engagement and behavior change found that while IT platform-based interventions had positive effects on health, there are few published reports of the engagement and usability within the interventions [50,55,56]. This study directly addresses these limitations.

Within fields of policy, and research alike, a priority is to develop interventions that are personalized, sustainable, and cost-effective. In the context of this, and widespread calls for research to address and alleviate health inequities [57,58], web-based interventions are promising because they can be tailored to the individual user, have potential for wide-scale use, and reach into rural settings to access hard to reach groups. For this reason, initiatives with a technology-based focus, in terms of delivery, may provide an efficient and cost-effective way of implementing large-scale health promotion, including PA. While concerns have been raised about the applicability of and access to internet interventions for older adults, population-based data suggest that in 2016, 92% of British Columbian adults used the internet and 75% of Canadians aged 55–64 used the internet every day [59]. In 2012, 86.5% of British Columbians had access to the internet at home, and it is highly likely that number has risen seven years later [60].

There is therefore good evidence to support the potential for the uptake of web-based interventions amongst older adults, and this potential will only increase with the growing use of and familiarity with the internet; in fact, it is likely that web-based information and support will continue to become even more compatible with the preferences and needs of successive cohorts as people age.

### 2.2. Development of the Intervention and Theoretical Framework

Following calls for PA intervention research to clearly explicate how theoretical components align with intervention components and study design, we now clarify the conceptual underpinnings of the intervention used for this study [49].

When considering how to design this intervention, we were guided by the multi-process action control (M-PAC) schematic [33], which builds from several streams of past theoretical literature. This framework is novel in part due to the explicit layering of motivational (e.g., attitude, perceived control), regulatory (e.g., planning), and reflective (e.g., habit, identity) processes that facilitate an initial intention into successful on-going behavior [33,61,62,63,64,65,66]. While traditional theories predicting PA behavior fixate upon the correlation of intention and PA, they are challenged by the enduring “intention behavior gap”, illustrated in research that has shown up to 48% of intenders failed to follow through with behavior [67]. The most recent Canadian PA “report card” cites that while 75% of adults can be classified as exercise “intenders” only 16% follow through [68]. Integrating reflexive processes such as habit and identity into conceptual models of PA participation has shown promising results thus far in research [45,65,66,68] and within the terrain of life transitions such as retirement and empty nest, and accounting for these processes may take on an even greater significance.

Recently our lab has ventured to operationalize the M-PAC model in a usable online platform, and a full description of methods and results for this process has been reported elsewhere [62]. This intervention is currently in use in our lab for new mothers and adolescents, and to date, one feasibility study has been completed with university students [69]. While some of these studies are still in progress, early results indicate that overall, university students and new mothers perceive proposed study procedures and the receipt of internet-administered support as acceptable [69]. These findings helped to inform the study procedures of the present feasibility study.

## 3. Methods

The study is being conducted in accordance with the Declaration of Helsinki, and was approved by the University of Victoria Human Research Ethics Board (HREB) on 18 September 2019. This protocol is reported according to the guidelines presented in the Consolidated Standards of Reporting Trials’ (CONSORT) 2010 statement extension for randomized pilot and feasibility studies [70]. The trial is registered with the Protocol Registration and Results System (PRS), Trial Number Register [NCT04116372].

### 3.1. Design

The study has a controlled baseline, post-intervention (10-weeks) evaluative design with an embedded qualitative and quantitative process evaluation. All intervention condition participants receive the internet-administered intervention for 10-weeks, comprising 10 modules addressing key behavioral concepts related to PA behavior, with additional content tailored towards empty nest and retired individuals. The development process of the base 10-week program has been reported elsewhere [62].

### 3.2. Eligibility Criteria

The eligible population includes recently retired and empty nest individuals, who are in addition not currently meeting PA guidelines (attaining less than 150 min of MVPA per week). We used the previous 6 months to delineate a split of “recent” or “not recently” retired, or had their last child leave the family home; however, we acknowledge that this split is relatively arbitrary and 12 months may encourage a larger reach of participants.

In other work exploring the habit discontinuity thesis, Verplanken et al. (2008) suggested that a period of 12 months may be suitable to evaluate the effects on travel mode use after moving home [39]. Later work broadly supports this concept, however, the strongest differences in the predicted probability were observed in the very early months after moving home, and soon decreased over time, suggesting a declining influence of contextual changes [71].

Exclusion criteria for our study were as follows: have not retired (self-defined) within 6 months or last child has not moved out of the familial home within 6 months. Participants are excluded if they are unable to speak and read English, and do not have access to the internet, a smart phone, or computer that can support the eHealth application we are using for the intervention. In addition, as assessed by the Get Active Questionnaire (GAQ) 2017, developed by the Canadian Society for Exercise Physiology (CSEP), a documented or patient-reported medical condition that would preclude participation, including within 6 months of having a diagnosis of/treatment for heart disease or stroke, or pain/discomfort/pressure in the chest during activities of daily living or during physical activity, a diagnosis of/treatment for high blood pressure, or a resting BP of 160/90 mmHg or higher, dizziness or lightheadedness during physical activity, shortness of breath at rest, loss of consciousness or fainting for any reason, concussion, pain or swelling in any part of the body (such as from an injury, acute flare-up of arthritis), or any other clinical condition that the person’s GP or clinician considers would make them unsuitable for participation in a physical activity program (such as diabetes, cancer, osteoporosis, asthma, spinal cord injury) [72]. Following recommendations of sample sizes of 50–60 being appropriate to assess the feasibility outcomes and estimate the sample size for a definite trial, we aim to recruit 80 participants [73].

### 3.3. Recruitment

To raise awareness of the study, advertisements have been placed on relevant social media sites and printed flyers have been distributed throughout the Greater Victoria Capital Regional District, in recreational centers, community centers, churches, grocery stores, coffee shops, libraries, and other relevant organizations. Study information has been circulated to volunteer groups, newcomers’ groups, as well as volunteer lists of potentially relevant organizations. Some outreach has been done in person at community events. Interested individuals receive more information about the study by telephoning or emailing the research coordinator.

Research suggests that reminders improves recruitment rates [74] and as such, where telephone numbers are provided, a member of the research team telephones all potential participants with the goal of answering any questions the person may have regarding the study and ascertaining interest in participation. In instances where a telephone call is not answered, a maximum of four additional telephone calls are made, and two telephone messages left over the following 2 weeks.

When potential participants interested in the study contact the research coordinator by email, they are provided with details of the study and asked for consent to make telephone contact to answer further study questions. At this time, they are verbally screened through the GAQ screening tool. Those who meet the eligibility criteria and are interested in study participation schedule a time to visit the lab.

### 3.4. Reasons for Non-Participation

Participants deciding to opt out of further contact are asked to provide further information and reasons for non-participation; this data will be used to determine possible barriers to participation and may provide information regarding the acceptability of the intervention. It is made clear to participations that the provision of reasons for non-participation is optional and participants are not required to report on why they do not wish to participate should they wish not to.

Participant flow through the study is illustrated in Figure 1 (adapted from CONSORT).

### 3.5. Enrollment and Randomization

Once a person passes the telephone screening process, a baseline appointment is booked at the University of Victoria Behavioural Medicine Lab, in which the participant is asked for informed written consent as well as to compete a hard copy of the GAQ. Once both forms are completed, a participant completes baseline measures, and is assigned a participant number for confidentiality. Participants are thereafter identified using participant numbers. Participants are subsequently randomized at a 1:1 ratio to either the waitlist or intervention group using simple sequentially randomized “sealed envelope” randomization codes. A research assistant not associated with the present study preformed the randomization sequence in Excel and prepared the opaque sealed envelopes to be distributed in a numerical sequence to confirmed eligible study participants. Following random assignment, “intervention condition” participants are immediately provided access to the online intervention and emailed a link to the web intervention. The web-based intervention proceeds over a 10-week period.

Waitlist-selected participants are told to resume any and all activities and goals they are pursuing. Waitlist-selected participants gain access to the online platform after completing the final questionnaire, completed 10 weeks following baseline.

The final questionnaire includes all measures for behavioral change and PA, and in addition, satisfaction and evaluation measures. An invitation for an exit interview is extended to the intervention group participants, and if willing, is scheduled in the following 10 days. Reminders are to be sent to participants one week preceding any requested questionnaire completion or check in calls.

### 3.6. Intervention

The online platform is an internet-administered, module learning-based intervention. Each module includes education alongside infographics, photos, text, audio, and video material. The intervention is designed to be delivered over a 10-week period, with participants encouraged to work at their own pace with the content. All modules are accessible to participants at any time to provide flexibility regarding completion. Online platform content is based on M-PAC (previously described) and is tailored towards the main barriers identified for people attempting to increase their PA levels as informed by previous research [33]. A brief overview of the module content is shown in Table 1, and linked to the M-PAC concepts as mapped by Rhodes (2017) [33]. The content on the platform intends to increase participant knowledge and awareness of the relevant PA behavioral change information, alongside the use of interactive online components, such as quizzes to enhance integration and understanding, and “action plans” to help participants make connections between the material and their lived experiences. The intervention has been developed with a scoping review of research, and popular material related to physical activity in transitions to empty nest or retirement in hopes to engage participants by including relevant examples that help participants relate the presented material to their lives.

### 3.7. Comparator Group

This feasibility study uses a waitlist control group; therefore, a proportion of participants experience a delay between enrolment and intervention. Participants allocated to the waiting control group receive access to the online platform 10 weeks after their baseline assessment. Participants are asked to complete their final survey at 10 weeks prior to receiving access to the online platform materials.

## 4. Data Collection

### 4.1. Primary Outcome Measures

The key feasibility outcomes examined via the proposed protocol concern the evaluation of study methods and procedures, including estimates of likely recruitment and retention rates, feasibility and acceptability of the intervention, and tests of the study procedures [75]. Following CONSORT, between group improvements regarding behavioral outcomes and efficacy are not examined at this stage. An embedded qualitative process evaluation will be used to examine:The acceptability of the intervention;Participants’ self-reported PA support needs;Potential mechanisms of change;The impact of the intervention.

Feasibility outcomes are shown in Table 2, adapted from Woodford et al. [76]. Progression criteria have been set a priori to facilitate a feasibility data analysis and provide guidance about whether to proceed to a future full-scale trial.

### 4.2. Baseline and Final Questionnaire

Data are collected by an online questionnaire via the University of Victoria’s institutional account with Survey Monkey, at baseline and post-intervention (10 weeks).

#### 4.2.1. Satisfaction and Usability

Study satisfaction is to be evaluated in the final questionnaire by a 6-item satisfaction and evaluation questionnaire, adapted from the satisfaction questionnaire from Forbes, Blanchard, Mummery, and Courneya (2015) [79]. Usability is be assessed by the 10-item System Usability Scale (SUS) [80].

#### 4.2.2. Sociodemographic Variables

Self-report data on participant age, gender, marital status, level of education, employment status, ethnicity, relationship status, number of children, ages of children, current housing situation, and experience using the internet are collected at the baseline interview.

### 4.3. Secondary Outcome Measures

Baseline questionnaires are completed in the lab and the 10-week questionnaires are sent as a link in an email and completed at the participant’s home. These questionnaires are designed to assess regular moderate to vigorous physical activity as well as other M-PAC constructs. The time frame of change for all measures is from baseline to 10 weeks and all are from the online questionnaire.

#### 4.3.1. Self-Reported Physical Activity

Participants self-report physical activity using the Godin Leisure-Time Exercise Questionnaire (GLTEQ) [81,82] at baseline and 10 weeks. The GLTEQ contains three questions, which assess the frequency of mild, moderate, and strenuous activity, during free time in a typical week. The questionnaire was modified using the results of Courneya and colleagues (2004) to an open duration [83]. In this format, we will calculate moderate and vigorous PA (MVPA) in minutes of physical activity by adding the multiplicative terms of the strenuous and moderate frequency × duration [84].

#### 4.3.2. M-PAC Constructs for Physical Activity

Constructs of affective attitude, instrumental attitude, and perceived behavioral control are assessed using the constructs of the theory of planned behavior [85]. Behavioral regulation is measured via items adapted from Sniehotta et al. [86] and Umstattd’s scale measuring physical activity self-regulation strategies in older adults [86]. Exercise identity (whether participants identify as being an exerciser) is measured via a modified exercise identity scale from Anderson and colleagues [1994, 1995, 1998]. Measures from these instruments have demonstrated excellent predictive validity and internal consistency in adult [61,87,88,89] samples, such as Self-Report Habit Index items from the measure developed by Verplanken and Orbell [90] and adapted to physical activity by Chatzisarantis and Hagger [91]. Finally, Anderson and Cychosz’s Exercise Identity Scale is used to measure participants’ identification as someone who participates in physical activity [92].

#### 4.3.3. Quality of Life

Psychosocial outcomes are primarily assessed using the short-form 12 (SF-12) health survey which measures health-related quality of life on a range of functional domains including vitality, social functioning, and overall well-being [93]. The SF-12 has been validated for adult populations with established evidence for reliability [94].

### 4.4. Post-Intervention Qualitative Interview

The process evaluation procedures involve a brief quantitative questionnaire to assess the use of the intervention material and overall satisfaction of the study. Intervention condition participants also have the option of participating in an in-person exit interview to explore the acceptability of the intervention and associated study procedures. In order to examine the possible mechanisms of change as well as suggestions for future intervention development and study procedures, participants’ views concerning the impact of the intervention are explored. The interview guide is semi-structured and includes 10 open-ended questions; it was informed by previous research examining the acceptability of online interventions and qualitative process evaluations that have proved useful in our prior evaluations [69]. To minimize bias, these interviews are facilitated by a research coordinator or assistant unaffiliated to the study. Audio recordings of interviews are to be transcribed verbatim and analyzed using a content and thematic analysis. All data are to be stored securely, and raw data are only to be accessible to the study chief investigator (RR) and researcher (AC).

### 4.5. Qualitative Analysis

Data from the exit interviews will be analyzed based on a thematic approach [95]. Research team members will read through the transcripts for familiarization, after which items of analytic interest will be identified, and an initial coding framework developed. Through repeated readings of the transcripts, a framework of patterns and concepts will be linked iteratively across interviews and refined though research memos and team meetings to discuss the emergent themes. For each theme, data extracts will be compiled on the basis of being representative and elaborating upon the research questions [96,97]. The reliability and validity of the analysis will be enhanced through an iterative data analysis, the use of a multi-method design, and the ongoing discussion of findings within the research team for scrutiny and feedback [97]. We will follow recommendations from previous literature [69,98] to reduce bias by analyzing the process data before the trial outcomes are known. The qualitative analysis procedure will be research question-oriented and anchored to the quantitative analysis [69,98].

## 5. Data Analyses

Data analyses will primarily be descriptive and address the outcomes relating to the feasibility of the intervention and study procedures. Progression criteria will be used to determine whether revisions should be considered before proceeding to a controlled trial.

### Quantitative Analyses

An adapted CONSORT diagram (Figure 1) for pilot and feasibility studies has been used to illustrate participant flow. Numbers of participants recruited by advertisements, flyers, in-person, etc., expressing initial interest, consented, assessed for eligibility, eligible, and included, are reported. Percentages of participants meeting the eligibility criteria of the participants assessed for eligibility, and participants eligible of the total number enrolled will be calculated. Reasons for ineligibility, ambiguities regarding eligibility criteria, and reasons for non-participation will be reported at each stage.

Follow-up rates will be calculated with 95% Confidence Intervals. Descriptive statistics for each outcome measurement at the eligibility interview, baseline and at the 10-week follow-up will be reported. Attrition proportions (for both intervention and study dropouts) will be reported.

Means, SDs, and frequencies for each portal activity relating to intervention adherence and use, including log-ins, opened modules and items, completed action plans, and use of optional support will be reported. Potential ambiguities regarding eligibility and other procedures will be tabulated into an Excel form and reported statistically. Types and numbers of measures undertaken and numbers of unforeseen safety issues, if any, will be reported.

## 6. Ethics

The study has been approved by the University of Victoria’s Human Research Ethical Review Board (HREB) in Victoria BC. Participant confidentiality is guaranteed. Informed consent is collected to ensure participants are aware of the conditions of the study participation. Participants are reminded of their rights to withdraw from the study without giving any reason. Participants are provided contact information within the study consent form for both the principal investigator, the study coordinator, and the independent UVIC HREB group should they have any cause for concern regarding the conduct of the trial. All participants are assigned a study code to de-identify the data and personal information about participants stored separately from the de-identified data. Data collected via the portal are to be stored on secure servers at University of Victoria, BC, Canada. All other data are stored in a locked filing cabinet, accessible only to the study personnel.

### Data Management and Confidentiality

Confidentiality procedures are outlined in the consent form and explicated during informed consent procedures conducted by the research coordinator at the baseline assessment. Each participant is provided with an identification number. Hard copies of any documentation are kept in a locked and secure environment (locked laboratory and cabinets) at the University of Victoria. Any data or personal information stored on computers are kept on a secure server. Questionnaire data are stored on SurveyMonkey servers in Canada.

## 7. Dissemination

The findings of the study and the subsequent proposal for the full-scale RCT will be presented to potential collaborators including local health authorities, governmental organizations, community organizations, and others through presentations at educational events. We will also disseminate the findings of the feasibility study and the proposed RCT to local, national, and international conferences as presentations and in scientific journals as published manuscripts. Participants who express interest in the study results will be made aware of any relevant publications. Public access to the participant level dataset will not be granted. There are no current plans to grant public access to the full protocol. All authors who have contributed to the protocol design are eligible for authorship on subsequent publications.

## 8. Results

At the time of submission (29 March 2020), we have obtained ethical approval, registered the trial, and recruited 31 participants. Recruitment is expected to be complete by the end of 2020. From the 31 participants recruited, 31 have completed the baseline measures, 19 have completed the 10-week measures, 19 have completed the study, and 2 have dropped out. See Figure 1 for the study procedures and participant flow chart.

## 9. Discussion

This protocol describes the implementation of a randomized controlled feasibility trial employing an online platform to increase physical activity behavior within recently empty nest or retired participants, based on the assumptions of the M-PAC framework as a conceptual model. Research findings will be important to public health as they may help to determine if providing low-cost, scalable, and evidence-based planning strategies for family physical activity can aid in producing higher adherence to physical activity.

## 10. Conclusions

The results of this feasibility study will inform the planning of a definitive RCT, by assessing the rates of recruitment, retention, adverse events, and data collection. The qualitative analysis will be used to further refine the empty nest/retirement-tailored online platform and the study protocol for the proposed trial. This randomized feasibility trial will build on PA promotion, habit discontinuity, and life-course research, evaluating whether an online PA intervention at the empty nest or retirement stage of life is an effective time point to promote PA. This study will also test the novel multi-process action control framework approach to PA-related behavioral change, exploring how reflexive processes of habit and identity may bridge adoption and maintenance in behavioral adherence.

## Figures and Tables

**Figure 1 ijerph-17-03544-f001:**
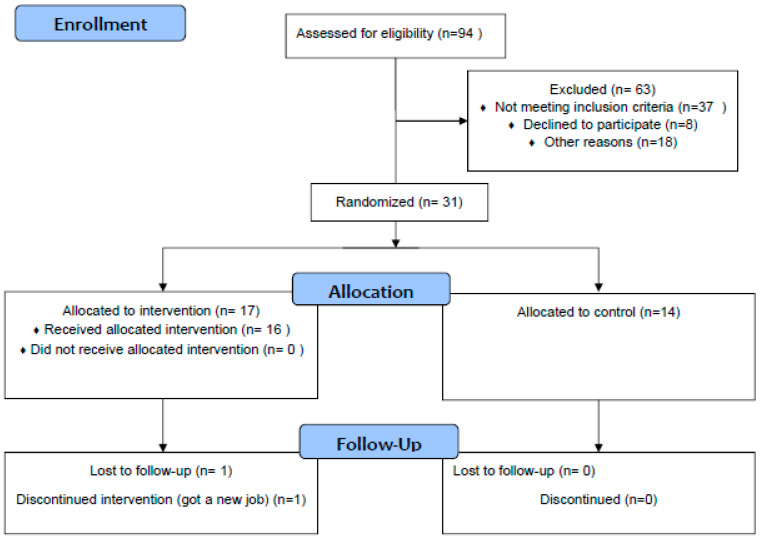
CONSORT 2010 Flow Diagram.

**Table 1 ijerph-17-03544-t001:** Overview of online platform lesson content linked to the theoretical/behavioral concepts used in this study.

	Title and Description	Corresponding Multi-Process Action Control Model Concept
Lesson 1	“Benefits of Physical Activity on Chronic Disease”	Initiating reflective processes (instrumental attitude, outcome expectations)
Lesson 2	“Mental benefits of Physical Activities”	Initiating reflective processes (instrumental attitude, outcome expectations)
Lesson 3	“Increasing Self Confidence for Physical Activities”	Initiating reflective processes (perceived capability)
Lesson 4	“Learning about your Emotion”	Ongoing reflective processes (affective attitude)
Lesson 5	“Building Social Support”	Ongoing reflective processes (affective attitude)
Lesson 6	“Building Physical Activity Opportunity”	Ongoing reflective processes (affective attitude)
Lesson 7	“Goal Setting and Planning”	Regulation processes Regulation
Lesson 8	“Self-Monitoring”	Regulation processes Regulation
Lesson 9	“Habit”	Regulation processes, reflexive processes (habit)
Lesson 10	“Identity”	Regulation processes, reflexive processes (identity)

**Table 2 ijerph-17-03544-t002:** Overview of feasibility outcomes and progression criteria, adapted from Woodford et al. (2018) [76].

Outcome	Evaluation Measures	Assessment Criteria for Progression
Recruitment and eligibility	Recruitment strategy	No criteria set
	Percentage assessed for eligibility; fulfilling inclusion criteria, and consented to participate (of total number screened)	1 day per week or 30 min MVPA change which equates to an approximate effect size of d = 0.35. This equates to roughly 65 per group (130 for a two-group trial). If we recruit for 2 years in the large trial (assuming a six-month time period for each person in the intervention) we need to be recruiting at least 6 per month. If we decide on a three armed trial, we will need to recruit at least 8 per month [77].
	Ambiguities regarding eligibility criteria	No criteria set
	Reasons for ineligibility	No criteria set
	Reasons for non-participation	No criteria set
Attrition	Rates of study dropout Rates of intervention dropout	70 > % 70 > % [78]
Resources needed to complete the study and the intervention	Length of time and cost required for: Study staff and resources required to administer the study Technical support required for online platform	No criteria set
Intervention Condition Participant Adherence	Number of: Opened modules, completed action plans Completed telephone check ins	70% > completing the introductory chapter, 10 modules and 2- and 5-week check in calls Full adherence to the intervention will be defined as: (1) completion of baseline meeting and questionnaire; (2) completion of all modules (3) completion of final questionnaire.
Participants’ use of the intervention/usage attrition	Number of: Log-ins Modules opened completed action plans/activities Time spent on each module	No criteria set
Participants’ acceptability and satisfaction with intervention	Reasons for withdrawal from study and intervention, reasons for not engaging with the online platform material.	No criteria set
	Participant feedback on the intervention material (including positive and negative) and of completing study procedures	70% > of participants using the intervention reporting that it is helpful 30% < participant reporting substantial negative consequences related to participation in the study and/or intervention [69].

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
