# Peer review of "Increasing Physical Activity in Empty Nest and Retired Populations Online: A Randomized Feasibility Trial Protocol"

_ijerph, 2020, doi:10.3390/ijerph17103544_

Round 1
Reviewer 1 Report
Dear author,
It is necessary to include in the abstract the main conclusion of the work.
Please read the article carefully
“Satisfaction with life, subjective well-being and functional skills in active older adults based on their level of physical activity practice”
you can enrich your introductory part as well as the article “Most valued components of the quality of life in older people than 60 years physically active” both can enrich all your text, when you talk about retired elderly people practicing physical activity. I consider it appropiate to inclurde these reference articles in your field of study that are recently published.
Reviewer 2 Report
IJERPH-780521
Summary
This is a protocol paper for a planned RCT (feasibility) targeting recent retirees and those who no longer have children living at home. The rationale for selecting these groups is that they represent significant periods of life transition where ‘habit strength’ may be weakened (and therefore behaviour change is more likely to occur). The lack of existing evidence for these populations also strengthens the rationale for this work. The paper is well written and adequate for a protocol paper. I have a few (mostly grammatical) suggestions below that the authors may wish to consider:
There are several cases where a space is missing after a period (L47, 58, etc).
L90: Please describe the ‘M-PAC’ acronym or avoid using it here as it has not been introduced.
L90–92: Why is a hypothesis presented for MVPA but no other primary or secondary objectives?
L103: Space before square bracket (and L113, L300, L301, etc.)
L115: Internet use doesn’t necessarily translate to ‘good evidence’ for ‘acceptability and relevance’ of web-based interventions. It shows the potential for uptake. The acceptability/relevance statement will need a reference, or simply rephrase the sentence.
L140: Is ibid necessary here given you are using numbered referencing?
L141–142: Suggest rewording this sentence as it is a little obtuse. Maybe “…helped to inform the procedures in the present study.”.
L160: Clarify whether ‘6 months’ is the upcoming 6 months, or the previous 6 months.
L161: Suggest semicolon (and 165)
L169: What does the word ‘currently’ mean here?
L229: Will the retired group and empty nest group receive a tailored intervention, or is the exact same material being delivered to both groups?
Table 2:
- The bullet points in are out of alignment which makes it unpleasant to read.
- Why has full adherence been deemed as completion of 5 (50%) of the models?
- Is it possible to assess the time spent on each module? This would be a useful metric to demonstrate participant time investment.
- Are the greater than signs ‘>’ on the wrong side of the number?
- There is a less than sign ‘<’ in the last assessment criteria box, but no corresponding number.
L276: Period, not a comma.
L286: Is it meant to read ‘from the online questionnaire.’?
L334: Why are 95% CIs calculated for missing items?
L335: Are post-treatment and 10-week follow-up the same thing? Or have I misunderstood?
L338–340: Reporting ambiguities doesn’t sound like quantitative analysis. Suggest moving out of this section.
Reviewer 3 Report
Considering the world pandemic I feel this is a very timely study for consideration. There are many people who may have recently retired or became empty nesters who could use a study like this to either improve or help them to maintain their physical activity levels.
There are some considerations for this study and the way that it is written. Because the study has yet to be completed, the tense of the manuscript should either be in future tense or remain in the present. For example 3.2. Eligibility criteria on page 4 line 158 is present tense but the line 168 it is past. There should be some continutity throughout the manuscript. Not having this made reading challegning.
Page 5 line 205 "Enrollment" is spelled incorrectly
Page 7 Table 2- the bullet points should be removed so the table is uniform
Page 8 Lines 287-290 More information should be available about this survey- how many questions? how is it scored?
Page 9 Lines 308-320 (4.4) More infomation about the interview guide is needed. Is it semi-structured? Will there be open and closed questions? How many questions are expected?
Page 9 A qualitative analysis is not provided.
Round 2
Reviewer 3 Report
The authors have done a fine job of addressing all the concerns of the initial document. I read through each of the reviews and commend them for putting forth the effort to make this manuscript more in depth and rich with information needed to conduct this study. Best wishes.